# The Prognostic Value of the CALLY Index in Sepsis: A Composite Biomarker Reflecting Inflammation, Nutrition, and Immunity

**DOI:** 10.3390/diagnostics15081026

**Published:** 2025-04-17

**Authors:** Ali Sarıdaş, Remzi Çetinkaya

**Affiliations:** 1Department of Emergency Medicine, University of Health Sciences, Prof. Dr. Cemil Taşçıoğlu City Hospital, 34383 Istanbul, Türkiye; 2Department of Emergency Medicine, University of Health Sciences, Gazi Yaşargil Training and Research Hospital, 21070 Diyarbakır, Türkiye; cetinkayaremzi2121@gmail.com

**Keywords:** albumin, C-reactive protein, lymphocyte, machine learning, mortality, sepsis

## Abstract

**Background/Objectives**: Sepsis remains a leading cause of mortality worldwide, necessitating the development of effective prognostic markers for early risk stratification. The C-reactive protein–albumin–lymphocyte (CALLY) index is a novel biomarker that integrates inflammatory, nutritional, and immunological parameters. This study aimed to evaluate the association between the CALLY index and 30-day all-cause mortality in sepsis patients. **Methods**: This retrospective cohort study included adult patients diagnosed with sepsis in the emergency department between 1 January 2022, and 1 January 2025. The CALLY index was calculated as (CRP × absolute lymphocyte count)/albumin. The primary outcome was 30-day all-cause mortality. Five machine learning models—extreme gradient boosting (XGBoost), multilayer perceptron, random forest, support vector machine, and generalized linear model—were developed for mortality prediction. Four feature selection strategies (gain score, SHAP values, Boruta, and LASSO regression) were used to evaluate predictor consistency. The clinical utility of the CALLY index was assessed using decision curve analysis (DCA). **Results**: A total of 1644 patients were included, of whom 345 (21.0%) died within 30 days. Among the five machine learning models, the XGBoost model achieved the highest performance (AUC: 0.995, R^2^: 0.867, MAE: 0.063, RMSE: 0.145). In gain-based feature selection, the CALLY index emerged as the top predictor (gain: 0.187), followed by serum lactate (0.185) and white blood cell count (0.117). The CALLY index also ranked second in SHAP analysis (mean value: 0.317) and first in Boruta importance (mean importance: 37.54). DCA showed the highest net clinical benefit of the CALLY index within the 0.10–0.15 risk threshold range. **Conclusions**: This study demonstrates that the CALLY index is a significant predictor of 30-day mortality in sepsis patients. Machine learning analysis further reinforced the prognostic value of the CALLY index.

## 1. Introduction

Sepsis is a life-threatening condition characterized by a dysregulated host response to infection, often leading to multiple organ failure and high mortality rates. Despite advancements in critical care and antimicrobial therapy, sepsis remains a major global health burden, affecting approximately 49 million individuals annually and causing nearly 11 million deaths, accounting for nearly 20% of all global fatalities [1,2,3]. In the United States alone, sepsis affects approximately 1.7 million adults annually and is responsible for over 250,000 deaths [4]. Hospital mortality rates range widely, from 10% to 52%, depending on disease severity, comorbidities, and the timing of interventions [5]. Given the high morbidity and mortality associated with sepsis, early risk stratification remains a key challenge in clinical management [6].

Several biomarkers have been explored for their prognostic value in sepsis, including C-reactive protein (CRP), albumin, and lymphocyte count. CRP is a widely recognized acute-phase reactant that is synthesized by the liver in response to inflammation. Elevated CRP levels have been associated with increased disease severity and worse outcomes in septic patients [7]. Albumin, a negative acute-phase protein, reflects both inflammatory and nutritional status. Hypoalbuminemia in sepsis is linked to capillary leakage, increased vascular permeability, and poor prognosis [8]. Lymphocyte count, on the other hand, serves as an indicator of immune competence. Sepsis-induced lymphopenia is a well-documented phenomenon that contributes to immune paralysis, increasing susceptibility to secondary infections and mortality [9].

While each of these markers provides valuable prognostic information independently, their combined assessment may offer a more comprehensive evaluation of a patient’s inflammatory and immune status. The C-reactive protein–albumin–lymphocyte index (CALLY index) has emerged as a novel biomarker incorporating inflammatory (CRP), nutritional (albumin), and immunological (lymphocyte) components to predict outcomes in critically ill patients [10]. Although the CALLY index has been investigated in conditions such as cancer, cardiovascular diseases, and postoperative infections, its prognostic significance in sepsis remains unclear [11,12].

This study aims to evaluate the association between the CALLY index and mortality in sepsis patients. We hypothesize that higher CALLY index values will be independently associated with increased short-term mortality. To explore this association, the study compares clinical and laboratory characteristics between survivors and non-survivors, and develops multiple machine learning models—including extreme gradient boosting, multilayer perceptron, random forest, support vector machine, and generalized linear model—to predict 30-day mortality. Different feature selection strategies are applied to assess the consistency of important predictors, and the clinical utility of the CALLY index is further evaluated using decision curve analysis. This study uniquely integrates a composite biomarker reflecting inflammation, nutrition, and immunity into multiple machine learning frameworks to predict 30-day mortality in sepsis. Its methodological design allows both performance comparison across algorithms and evaluation of clinical utility, offering a data-driven perspective on early risk stratification in this high-risk population.

## 2. Materials and Methods

### 2.1. Study Design

This retrospective cohort study investigated the association between the CALLY index and 30-day all-cause mortality in patients diagnosed with sepsis. Data were extracted from the electronic health records (EHRs) of Diyarbakir Gazi Yaşargil Education and Research Hospital, a tertiary care center, between 1 January 2022, and 1 January 2025. The study adhered to the Strengthening the Reporting of Observational Studies in Epidemiology (STROBE) guidelines and was approved by the Diyarbakir Gazi Yaşargil Education and Research Hospital Ethics Committee (approval number: 326, date: 7 February 2025), in accordance with the Declaration of Helsinki [13]. Due to the retrospective nature of the study, the requirement for informed consent was waived. Sepsis was defined using the Sepsis-3 criteria, which characterize sepsis as life-threatening organ dysfunction due to a dysregulated host response to infection. Organ dysfunction was determined based on an increase in the Sequential Organ Failure Assessment (SOFA) score of ≥2 points from baseline [14].

### 2.2. Study Population

The study included adult patients (≥18 years old) who presented to the emergency department of a tertiary care hospital and were diagnosed with sepsis between 1 January 2022 and 1 January 2025. Sepsis cases were identified through EHRs. Patients were excluded if they had end-stage renal disease requiring dialysis, acute kidney injury at admission, or chronic liver disease, including cirrhosis, as these conditions could significantly affect albumin levels and inflammatory responses. Additionally, patients with severe malnutrition or recent major surgery were not included, as these factors could distort inflammatory markers and albumin levels. Patients who were transferred from another hospital were excluded to ensure standardized data collection and avoid incomplete clinical information. Finally, those with missing laboratory values necessary for calculating the CALLY index were also excluded from the study.

### 2.3. Data Collection and Variable Definitions

Patient data were extracted from the EMRs. The collected variables included demographic characteristics, clinical presentation, laboratory findings, and patient outcomes. Demographic data included age, sex, and body mass index (BMI). Clinical data included comorbid conditions such as hypertension, diabetes, coronary artery disease, chronic obstructive pulmonary disease, and chronic kidney disease. Vital signs recorded at admission included heart rate, respiratory rate, systolic and diastolic blood pressure, oxygen saturation, and Glasgow Coma Scale score. Laboratory parameters obtained at the time of emergency department presentation included white blood cell count, neutrophil count, lymphocyte count, CRP, albumin, procalcitonin, blood urea nitrogen (BUN), creatinine, lactate, total bilirubin, aspartate aminotransferase (AST), alanine aminotransferase (ALT), and arterial blood gas values.

The CALLY index was calculated as the product of CRP and absolute lymphocyte count, divided by albumin. CRP levels were expressed in milligrams per liter (mg/L), albumin in grams per deciliter (g/dL), and absolute lymphocyte count in units of 109/L. To ensure data accuracy and minimize bias, data extraction was conducted by two independent researchers who underwent standardized training prior to the study. Any inconsistencies in recorded values were reviewed and resolved by a third investigator. Quality control procedures included systematic reviews for missing or implausible values, with inconsistent records cross-checked against the original hospital database.

### 2.4. Outcome

The primary outcome of this study was 30-day all-cause mortality, defined as death occurring within 30 days of hospital admission. Mortality data were obtained from electronic medical records and verified using national death registries.

### 2.5. Machine Learning Modeling

To construct predictive models for 30-day mortality in sepsis, five supervised machine learning algorithms were implemented: extreme gradient boosting (XGBoost), multilayer perceptron (MLP), random forest, support vector machine (SVM), and generalized linear model (GLM). These algorithms were selected to encompass a diverse range of model architectures—boosted decision trees, neural networks, ensemble bagging, kernel-based classifiers, and linear models—to enable a comprehensive comparison across different methodological frameworks. XGBoost was included due to its demonstrated superiority in sepsis prognostication, particularly in handling missing data, capturing nonlinear relationships, and minimizing overfitting through built-in regularization. Prior studies have shown that XGBoost effectively models complex clinical datasets and achieves strong discriminative performance in sepsis-related mortality prediction [15,16]. MLP was selected as a deep learning approach capable of capturing higher-order interactions that may not be linearly separable. Although more sensitive to overfitting, its inclusion provided insight into the behavior of multilayer neural networks in this context. Random forest was chosen for its robustness and inherent feature importance estimation, which has been previously leveraged in sepsis datasets for variable ranking and prediction [17]. SVM was incorporated as a classical algorithm known to perform well in high-dimensional spaces, especially when margin-based separation is feasible. Finally, GLM was included as a transparent, interpretable model commonly used in biomedical research, serving as a comparative benchmark against more complex models. The inclusion of these five algorithms allowed for evaluation of the relative advantages and limitations of each modeling approach in sepsis-related mortality prediction, while minimizing algorithmic bias. Subsequent comparisons of predictive performance and feature selection outcomes were conducted to guide model selection.

### 2.6. Data Analysis

Continuous variables were assessed for normality using the Kolmogorov–Smirnov test and visual inspection of histograms. Normally distributed variables were reported as mean ± standard deviation (SD) and compared using the independent *t*-test, while non-normally distributed variables were presented as median [interquartile range (IQR)] and compared using the Mann–Whitney U test. Categorical variables were expressed as counts (percentages) and analyzed using the chi-square test or Fisher’s exact test, as appropriate. To develop predictive models for sepsis-related mortality, we implemented five machine learning algorithms: XGBoost, MLP, random forest, SVM, and GLM. The dataset was randomly divided into training (80%) and testing (20%) subsets, stratified by mortality outcome to preserve class balance. All model training, hyperparameter tuning, and performance evaluation were conducted on the training dataset, with test performance assessed independently to avoid data leakage. The predictive performance of each model was evaluated using area under the receiver operating characteristic curve (AUC), coefficient of determination (R^2^), mean absolute error (MAE), and root mean square error (RMSE). AUC comparisons between models were performed using DeLong’s test for correlated ROC curves. After identifying XGBoost as the best-performing model based on test set metrics, we conducted additional analyses using four different feature selection strategies—gain score, SHAP values, Boruta, and LASSO regression—to explore the impact of variable selection on model performance. Feature importance for the gain-based model was assessed using the gain metric, which quantifies the relative contribution of each predictor to the overall model. Model performance for each feature selection approach was compared using the same evaluation metrics on the test set. To assess the clinical utility of the CALLY index, decision curve analysis was performed across a range of probability thresholds. Net benefit was calculated by comparing true positive and false positive rates, with comparisons against treat-all and treat-none strategies. The probability threshold at which the CALLY index provided the highest net benefit was identified as the optimal range for clinical decision-making. All statistical analyses were conducted using R software (version 4.4.2; R Foundation for Statistical Computing, Vienna, Austria). A two-tailed *p*-value < 0.05 was considered statistically significant.

## 3. Results

A total of 1644 patients were included, with 345 (21.0%) classified as deceased (Figure 1). Deceased patients were older (*p* < 0.001) and had a higher prevalence of chronic kidney disease (*p* = 0.013), whereas other comorbidities such as hypertension, diabetes, coronary artery disease, and chronic obstructive pulmonary disease did not show statistically significant differences between groups (Table 1). In terms of vital signs, deceased patients had lower systolic and diastolic blood pressure (*p* = 0.001, *p* < 0.001, respectively), higher heart rate (*p* < 0.001), higher respiratory rate (*p* < 0.001), lower oxygen saturation (*p* < 0.001), and worse neurological status, as indicated by lower Glasgow Coma Scale scores (*p* < 0.001) (Table 1).

Laboratory parameters revealed significantly higher inflammatory markers in deceased patients, including higher white blood cell count, neutrophil count, procalcitonin, and C-reactive protein levels (all *p* < 0.001), while lymphocyte count and albumin levels were significantly lower (both *p* < 0.001). Additionally, lactate, blood urea nitrogen, total bilirubin, aspartate aminotransferase, and alanine aminotransferase were markedly higher in non-survivors (all *p* < 0.001) (Table 2). The CALLY index was significantly higher in deceased patients (*p* < 0.001), suggesting its potential role as a predictor of sepsis-related mortality (Table 2). The CALLY index was significantly elevated in the deceased group (72.4 [23.3–190] vs. 24.3 [15.9–34.6], *p* < 0.001), supporting its potential as a prognostic marker for sepsis-related mortality.

To assess the prognostic utility of the CALLY index alongside other clinical parameters, we developed multiple machine learning models to predict 30-day sepsis-related mortality. Specifically, we compared the performance of five different algorithms: XGBoost, MLP, random forest, SVM, and GLM. Each model was trained and evaluated using the same dataset, and their predictive performance was assessed on the test set using metrics including the area under the ROC curve (AUC), R^2^, MAE, and RMSE. Summary metrics are presented in Table 3. The gain-based XGBoost model achieved the highest test performance among all machine learning models, with an AUC of 0.995 (95% CI: 0.991–1.000), R^2^ of 0.867, MAE of 0.063, and RMSE of 0.145. To evaluate its statistical superiority, we compared it against MLP, random forest, SVM, and GLM using DeLong’s test. XGBoost significantly outperformed all other models in terms of AUC (*p* < 0.001 for each pairwise comparison), confirming its superior discriminative power for sepsis-related mortality prediction.

We trained four distinct XGBoost models using features selected by different methods: gain score, SHAP values, Boruta, and LASSO. The top-ranked features identified by each method are summarized in Table 4, demonstrating considerable overlap across approaches (Figure 2). Notably, the CALLY index consistently ranked among the most influential predictors—first in the gain-based model, second in the SHAP-based model, and the top feature in Boruta’s importance scores—while it was excluded by LASSO due to its penalization of multicollinearity and composite features. To evaluate the impact of feature selection strategy, we trained and tested models using each subset of selected features. The gain-based model achieved an AUC of 0.99517 (95% CI: 0.99071–0.99964), with R^2^ = 0.867, MAE = 0.063, and RMSE = 0.145 on the test dataset. The training metrics were R^2^ = 0.973, MAE = 0.037, and RMSE = 0.076. The SHAP-based model achieved an AUC of 0.99712 (95% CI: 0.99468–0.99956), with R^2^ = 0.879, MAE = 0.063, and RMSE = 0.140 on the test dataset. The training metrics were R^2^ = 0.970, MAE = 0.038, and RMSE = 0.079. The Boruta-based model achieved an AUC of 0.99843 (95% CI: 0.997–1.000), with R^2^ = 0.904, MAE = 0.059, and RMSE = 0.128 on the test dataset. The training metrics were R^2^ = 0.983, MAE = 0.033, and RMSE = 0.062. The LASSO-based model achieved an AUC of 0.99723 (95% CI: 0.995–1.000), with R^2^ = 0.881, MAE = 0.066, and RMSE = 0.141 on the test dataset. The training metrics were R^2^ = 0.978, MAE = 0.036, and RMSE = 0.070. To statistically compare the discrimination performance of these models, we applied DeLong’s test for correlated ROC curves. There was no statistically significant difference in AUC between the gain- and SHAP-based models (*p* = 0.1675), gain and Boruta (*p* = 0.0865), or gain and LASSO (*p* = 0.3288). Similarly, the SHAP-based model did not differ significantly from Boruta (*p* = 0.1043) or LASSO (*p* = 0.9326). The Boruta- and LASSO-based models also did not differ significantly (*p* = 0.2480).

Decision curve analysis demonstrated that the CALLY index provided the highest net benefit within the 0.10–0.15 risk threshold range, indicating its optimal clinical utility in predicting sepsis-related mortality (Table 5, Figure 3). At a 0.05 threshold, the CALLY index yielded a net benefit of 0.202, which was superior to both the treat-all (net benefit: 0.19) and treat-none strategies (net benefit: 0.00). The highest observed clinical utility was at 0.10 (net benefit: 0.12), after which a gradual decline in net benefit was noted. Beyond a 0.20 threshold, the net benefit decreased progressively, with only marginal additional value at 0.25 (net benefit: 0.01). At thresholds of 0.30 and above, the benefit diminished substantially (net benefit: 0.00), indicating that risk stratification at these levels may not provide meaningful clinical guidance. At 0.35 and beyond, the CALLY index did not offer additional predictive value over treating all patients or no patients.

## 4. Discussion

This study investigated the association between the CALLY index and 30-day all-cause mortality in sepsis patients. Our findings demonstrate that the CALLY index is significantly elevated in non-survivors, supporting its potential role as a prognostic biomarker in sepsis. Additionally, machine learning-based predictive modeling identified CALLY as one of the most influential features for sepsis mortality, alongside lactate, white blood cell count, and neutrophil count. These results highlight the importance of incorporating inflammatory, nutritional, and immunological markers into sepsis risk stratification models.

CRP is a widely recognized acute-phase reactant synthesized by the liver in response to systemic inflammation and infection. It plays a crucial role in host defense mechanisms by activating complement pathways and promoting phagocytosis. In sepsis, CRP levels can increase significantly, often correlating with disease severity and systemic inflammatory response [18]. However, despite its routine clinical use, the prognostic value of CRP in predicting sepsis outcomes remains controversial. In this study, CRP levels were significantly higher in non-survivors compared to survivors, suggesting a potential association with sepsis-related mortality. However, when incorporated into the CALLY index, its predictive value appeared more robust. This finding aligns with prior research indicating that single biomarkers like CRP may have limited predictive accuracy for mortality, while composite indices incorporating multiple physiological parameters may enhance risk stratification. The recent literature presents conflicting findings regarding the prognostic utility of CRP. A study by Schupp et al. demonstrated that while CRP levels were elevated in sepsis and septic shock patients, their individual predictive value for 30-day mortality was poor [19]. Similarly, Tian et al. found that although CRP was significantly higher in non-survivors, its prognostic power improved when combined with other inflammatory markers such as procalcitonin and neutrophil-to-lymphocyte ratio [20]. In contrast, a systematic review by Huang et al. suggested that CRP, when combined with other biomarkers like monocyte distribution width, could enhance early sepsis detection but remained suboptimal as a stand-alone prognostic marker [21].

Albumin, the second component of the CALLY index, is a crucial negative acute-phase protein that plays a key role in maintaining oncotic pressure, regulating vascular permeability, and modulating inflammatory responses. In sepsis, hypoalbuminemia is commonly observed due to increased capillary leakage, hepatic dysfunction, and protein loss, often correlating with worse clinical outcomes [22]. In this study, albumin levels were significantly lower in non-survivors compared to survivors, reinforcing its potential role as a prognostic marker in sepsis. Patients with hypoalbuminemia exhibited increased inflammatory markers and higher mortality, consistent with previous research indicating that low albumin levels are associated with poor prognosis in critically ill patients. Recent studies further highlight the prognostic significance of hypoalbuminemia in sepsis. Kumar et al. demonstrated that sepsis patients with hypoalbuminemia had a significantly higher mortality rate (29.3%) compared to those with normal albumin levels (11.4%) [23]. Similarly, Furukawa et al. found that hypoalbuminemia (<2.8 mg/dL) independently predicted mortality in sepsis patients [24].

Lymphocytes, the third component of the CALLY index, play a critical role in immune regulation and host defense against infections. Sepsis-induced lymphopenia is a well-documented phenomenon that contributes to immune suppression, increased secondary infections, and higher mortality rates. This immune dysregulation leads to an imbalance between pro-inflammatory and anti-inflammatory responses, predisposing patients to poorer outcomes [25]. In this study, lymphocyte counts were significantly lower in non-survivors compared to survivors, reinforcing the association between lymphopenia and sepsis-related mortality. Prior studies have demonstrated that persistent lymphopenia is an independent predictor of poor prognosis in septic patients. Cilloniz et al. found that lymphopenia was independently associated with increased ICU admission rates and 30-day mortality in patients with community-acquired pneumonia and sepsis [26]. Similarly, Vahedi et al. reported that sepsis patients with lymphopenia had significantly higher SOFA scores, greater ICU admission rates, and increased risk of 28-day mortality [27].

The CALLY index, a composite biomarker integrating inflammatory (CRP), nutritional (albumin), and immunological (lymphocyte) components, has gained attention for its potential role in predicting outcomes in critically ill patients. Our study found that the CALLY index was significantly lower in non-survivors compared to survivors, indicating a possible association with sepsis-related mortality. Patients with lower CALLY index values had higher inflammatory markers, worse organ function, and increased mortality, suggesting that a weakened inflammatory, nutritional, and immune response contributes to poor prognosis. There is still limited research on the CALLY index in sepsis, making our findings particularly relevant. A recent study by Zhang et al. assessed the prognostic value of the CALLY index in ICU patients with sepsis, reporting that higher CALLY index values correlated with significantly lower 30-day and 60-day mortality rates [28]. Their findings are in agreement with our results, further indicating that the CALLY index may serve as a useful tool for mortality risk assessment in sepsis. Future research should evaluate whether integrating the CALLY index into existing sepsis scoring systems improves mortality prediction and clinical decision-making.

In this study, we employed machine learning-based feature selection (XGBoost) to identify the most influential predictors of sepsis-related mortality. The CALLY index emerged as one of the top-ranked variables, alongside lactate, white blood cell count, and neutrophil count, suggesting that a combined inflammatory, nutritional, and immunological marker provides valuable prognostic information. Traditional risk models in sepsis, such as SOFA and qSOFA, primarily rely on organ dysfunction and hemodynamic parameters, whereas the CALLY index integrates systemic inflammation, protein metabolism, and immune suppression, which may explain its predictive strength. Prior machine learning models have primarily relied on laboratory-based scoring systems, such as blood count parameters and traditional sepsis biomarkers [29,30]. However, our findings suggest that incorporating the CALLY index, which integrates immune–nutritional interactions, may enhance sepsis mortality prediction by capturing a broader range of physiological disturbances.

Additionally, decision curve analysis demonstrated that the CALLY index provided the highest net benefit within the 0.10–0.15 risk threshold range, suggesting a potential role in early risk stratification. This means that, within this probability range, the CALLY index adds predictive value over treat-all or treat-none strategies, supporting its use in guiding clinical decision-making. Early identification of high-risk patients using a composite index like CALLY may allow for timely escalation of care, resource optimization, and improved triage decisions in sepsis management. Future studies should explore whether integrating the CALLY index into established sepsis scoring models further improves clinical utility and decision-making accuracy.

### Limitations

This study has several limitations that should be acknowledged. First, as a retrospective, single-center study, the findings may be subject to selection bias and may not be fully generalizable to other healthcare settings or populations. Second, while the CALLY index integrates inflammatory, nutritional, and immunological parameters, it does not account for other potential confounders, such as dynamic changes in biomarker levels over time or the impact of sepsis treatment strategies. Third, although we used a machine learning-based predictive model to assess the importance of CALLY in mortality prediction, external validation in independent cohorts is required to confirm its robustness. Finally, the study relied on electronic health records, which, despite systematic data verification, may still be affected by missing or misclassified data.

## 5. Conclusions

In this retrospective cohort study, we demonstrated that the CALLY index is significantly associated with 30-day all-cause mortality in sepsis patients, with higher values observed in non-survivors. Furthermore, machine learning-based analysis identified CALLY as one of the most influential predictors of mortality, emphasizing its potential role in sepsis risk stratification. These findings suggest that incorporating inflammatory, nutritional, and immunological markers into clinical decision-making may enhance early prognostic assessment in sepsis. However, further prospective multicenter studies are needed to validate these results and determine the optimal clinical application of the CALLY index in sepsis management.

## Figures and Tables

**Figure 1 diagnostics-15-01026-f001:**
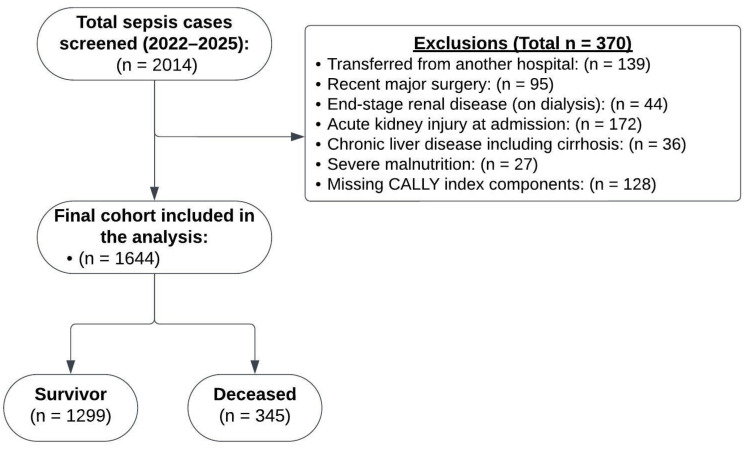
Patient flowchart.

**Figure 2 diagnostics-15-01026-f002:**
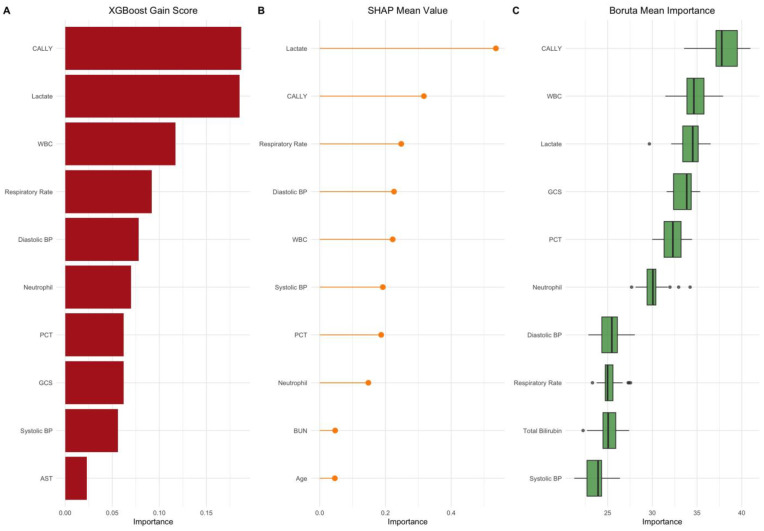
Comparison of CALLY index importance across three feature selection methods. (**A**) Gain score from XGBoost model. (**B**) SHAP mean value showing the average impact on model prediction. (**C**) Boruta importance scores with variability across iterations.

**Figure 3 diagnostics-15-01026-f003:**
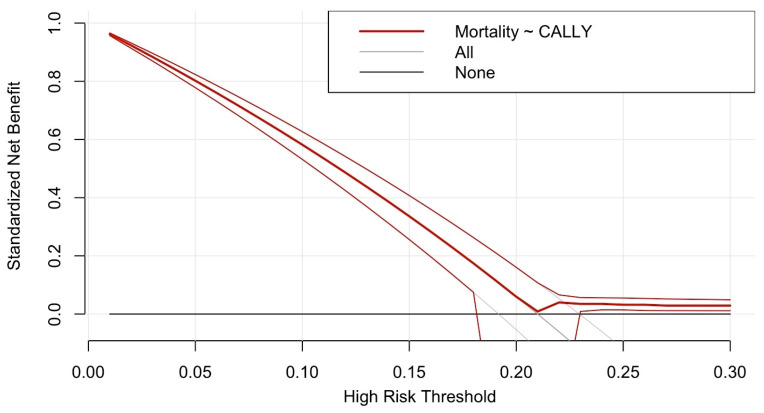
Decision curve analysis for the CALLY index in predicting sepsis mortality.

**Table 1 diagnostics-15-01026-t001:** Patient characteristics, comorbidities, and vital signs.

Variable	Survivor (*n* = 1299)	Deceased (*n* = 345)	*p*	Mean Difference (95% CI)
Age (years)	62.0 ± 12.0	71.2 ± 9.6	<0.001	9.8 (7.91–10.34)
Sex (Male, *n*%)	726 (55.9%)	180 (52.2%)	0.235	-
Hypertension, *n* (%)	767 (59.1%)	219 (63.5%)	0.157	-
Diabetes, *n* (%)	845 (65.1%)	224 (64.9%)	1.000	-
CKD, *n* (%)	301 (23.2%)	103 (29.9%)	0.013	-
CAD, *n* (%)	397 (30.6%)	120 (34.8%)	0.154	-
COPD, *n* (%)	254 (19.6%)	75 (21.7%)	0.412	-
Malignancy, *n* (%)	206 (15.9%)	38 (11.0%)	0.030	-
Dementia, *n* (%)	170 (13.1%)	44 (12.8%)	0.937	-
Multiple Infections, *n* (%)	197 (15.2%)	63 (18.3%)	0.190	-
BMI, kg/m^2^)	27.6 ± 3.0	27.8 ± 3.2	0.243	-
Systolic BP (mmHg)	95.9 ± 9.2	94.2 ± 8.0	0.001	1.7 (0.74–2.71)
Diastolic BP (mmHg)	60.5 ± 8.0	51.8 ± 5.7	<0.001	8.7 (7.92–9.40)
Heart Rate (bpm)	98.2 ± 15.2	110.3 ± 13.0	<0.001	12.1 (1.46–13.66)
Respiratory Rate (breaths/min)	21.3 ± 5.1	27.3 ± 3.9	<0.001	6 (5.48–6.47)
Temperature (°C)	37.8 ± 0.5	38.0 ± 0.5	<0.001	0.2 (0.18–0.29)
Oxygen Saturation (%)	94.2 ± 3.0	91.2 ± 3.9	<0.001	3 (2.6–3.49)
Glasgow Coma Scale	12.2 ± 2.9	9.0 ± 3.8	<0.001	3.2 (2.74–3.60)
qSOFA Score	2.1 ± 0.8	2.2 ± 0.8	0.243	-
SOFA Score	12.1 ± 7.1	14.9 ± 7.3	<0.001	2.8 (1.92–3.65)

CKD: Chronic Kidney Disease; CAD: Coronary Artery Disease; COPD: Chronic Obstructive Pulmonary Disease; BMI: Body Mass Index; BP: Blood Pressure; SOFA: Sequential Organ Failure Assessment; qSOFA: Quick Sequential Organ Failure Assessment; CI: Confidence Interval.

**Table 2 diagnostics-15-01026-t002:** Laboratory parameters and sepsis biomarkers.

Variable	Survivor (*n* = 1299)	Deceased (*n* = 345)	*p*	Mean Difference (95% CI)
White Blood Cell Count (×10^9^/L)	13.7 ± 2.3	17.4 ± 3.1	<0.001	3.7 (3.35–4.06)
Neutrophil Count (×10^9^/L)	10.4 ± 2.1	13.2 ± 3.3	<0.001	2.8 (2.43–3.16)
Lymphocyte Count (×10^9^/L)	1.2 ± 0.6	0.7 ± 0.7	<0.001	0.5 (0.39–0.54)
Platelet Count (×10^3^/µL)	281.4 ± 203.8	279.9 ± 189.1	0.901	-
Procalcitonin (ng/mL)	4.1 ± 2.0	7.3 ± 3.0	<0.001	3.2 (2.9–3.58)
C-reactive Protein (mg/L)	129.7 [111–150.2]	183.7 [127.4–241.2]	<0.001	-
Albumin (g/dL)	3.1 ± 0.5	2.4 ± 0.9	<0.001	0.7 (0.56–0.77)
Lactate (mmol/L)	2.6 ± 1.7	4.3 ± 1.9	<0.001	1.7 (1.45–1.88)
Blood Urea Nitrogen (mg/dL)	11.1 ± 3.5	13.9 ± 3.4	<0.001	2.8 (2.43–3.26)
Creatinine (mg/dL)	1.54 ± 0.37	1.59 ± 0.33	0.018	0.05 (0.01–0.90)
Total Bilirubin (mg/dL)	1.5 ± 1.3	2.3 ± 1.6	<0.001	0.8 (0.64–1.02)
Aspartate Aminotransferase (U/L)	63.2 ± 48.5	93.3 ± 56.1	<0.001	30.1 (23.6–36.6)
Alanine Aminotransferase (U/L)	52.1 ± 43.0	69.5 ± 51.5	<0.001	17.4 (11.5–23.3)
Glucose (mg/dL)	192.7 ± 143.2	223.2 ± 166.7	0.002	30.5 (11.2–49.7)
Sodium (mEq/L)	140.4 ± 17.9	141.7 ± 17.7	0.216	-
Potassium (mEq/L)	4.8 ± 1.8	4.6 ± 1.8	0.152	-
Bicarbonate (mEq/L)	20.6 ± 11.5	21.7 ± 11.6	0.136	-
Cally Index	24.3 [15.9–34.6]	72.4 [23.3–190]	<0.001	-

CI: Confidence interval.

**Table 3 diagnostics-15-01026-t003:** Performance metrics of machine learning models in predicting sepsis mortality.

Model	Dataset	AUC (95% CI)	R^2^	MAE	RMSE
XGBoost (Gain)	Test	0.995 (0.991–1.000)	0.867	0.063	0.145
	Train	0.996 (0.992–1.000)	0.973	0.037	0.076
MLP	Test	0.988 (0.980–0.996)	0.772	1.033	1.053
	Train	0.994 (0.991–0.998)	0.878	1.033	1.043
Random Forest	Test	0.991 (0.987–0.993)	0.851	1.011	1.024
	Train	0.994 (0.990–0.098)	0.982	1.000	1.002
SVM	Test	0.992 (0.987–0.997)	0.810	1.006	1.021
	Train	0.993 (0.988–0.997)	0.862	0.999	1.010
GLM	Test	0.991 (0.985–0.997)	0.810	1.008	1.024
	Train	0.993 (0.989–0.997)	0.859	1.000	1.012

AUC = Area Under the Receiver Operating Characteristic Curve; CI = Confidence Interval; R^2^ = Coefficient of Determination; MAE = Mean Absolute Error; RMSE = Root Mean Square Error; MLP = Multilayer Perceptron; SVM = Support Vector Machine; GLM = Generalized Linear Model.

**Table 4 diagnostics-15-01026-t004:** Comparison of feature importance scores across different selection methods.

Rank	Feature	XGBoost Gain Score	SHAP Mean Value	Boruta Mean Importance	LASSO Coefficient
1	CALLY Index	0.187	0.317	37.54	–
2	Serum Lactate (mmol/L)	0.185	0.536	34.16	0.417
3	White Blood Cell Count	0.117	0.222	35.07	0.394
4	Respiratory Rate (breaths/min)	0.092	0.248	25.63	0.256
5	Diastolic Blood Pressure (mmHg)	0.078	0.226	25.88	–0.121
6	Neutrophil Count (10^9^/L)	0.070	0.148	30.17	0.208
7	Procalcitonin (ng/mL)	0.062	0.187	32.17	6.462
8	Glasgow Coma Scale	0.062	0.099	33.25	–0.226
9	Systolic Blood Pressure (mmHg)	0.056	0.192	23.66	0.106
10	Aspartate Aminotransferase (U/L)	0.023	0.055	22.09	0.009
11	Total Bilirubin (mg/dL)	0.022	0.032	24.87	0.329
12	Age (years)	0.012	0.046	14.44	0.068
13	SOFA Score	0.011	0.012	21.65	–
14	Blood Urea Nitrogen (mg/dL)	0.011	0.047	14.56	0.239
15	Oxygen Saturation (%)	0.008	0.037	19.38	–0.212

**Table 5 diagnostics-15-01026-t005:** Decision curve analysis metrics for the CALLY index.

Threshold	Net Benefit (CALLY Index)	Net Benefit (Treat All)	Net Benefit (Treat None)	Clinical Interpretation
0.05	0.202	0.19	0.00	CALLY index provides a moderate net benefit at very low-risk thresholds.
0.10	0.12	0.10	0.00	The highest clinical utility observed within this range.
0.15	0.07	0.05	0.00	CALLY remains superior to treat-all and treat-none approaches.
0.20	0.03	0.02	0.00	Declining net benefit, but still clinically useful.
0.25	0.01	0.00	0.00	Marginal benefit beyond this threshold.
0.30	0.00	0.02	0.00	The benefit diminishes at higher thresholds.
0.35+	0.00	0.00	0.00	CALLY index offers no additional benefit beyond this threshold.

## Data Availability

The data supporting the findings of this study are available upon reasonable request from the corresponding author. Due to privacy and ethical restrictions, the dataset cannot be publicly shared.

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
