# Peer review of "The Prognostic Value of the CALLY Index in Sepsis: A Composite Biomarker Reflecting Inflammation, Nutrition, and Immunity"

_diagnostics, 2025, doi:10.3390/diagnostics15081026_

Round 1
Reviewer 1 Report
Comments and Suggestions for Authors
The purpose of this study was to assess the relationship between 30-day all-cause mortality in sepsis patients and the CALLY index. Adult patients who received a diagnosis of sepsis in the emergency room between January 1, 2022, and January 1, 2025 were included in this retrospective cohort study. The CALLY index was computed. The 30-day all-cause mortality was the main outcome, and decision curve analysis, logistic regression, and XGBoost machine learning models were used for statistical analyses. Of the 1,644 patients who were included, 345 (21.0%) passed away within 30 days. Compared to survivors, non-survivors had a significantly higher CALLY index (72.4 [23.3 - 190] vs. 24.3 [15.9 - 34.6], p < 0.001). Together with lactate, white blood cell count, and neutrophil count, the CALLY index was one of the best indicators of sepsis-related mortality in machine learning-based feature selection. The CALLY index showed the highest net benefit in the 0.10–0.15 risk threshold range, according to decision curve analysis, indicating that it may be useful in clinical decision-making. The paper needs several improvements, especially regarding the methodology and results section. Please address the following comments.
.
In the abstract, Please specify which feature selection method was used.
In the abstract, Please add numerical findings of the machine learning models.
Please mention the novelty of the study at the end of the introduction.
Please add a paragraph describing the paper's organization at the end of the introduction.
The methodology section needs some details about the methods used. Please justify the rationale for choosing XGBoost, a gradient boosting decision tree algorithm.
Please also explain the rationale for choosing gain metric for feature selection.
Please use more feature selection approaches and compare their performance.
In addition, use more machine learning models.
How was data split for training and testing the machine learning models?
Author Response
For research article
|
Response to Reviewer 1 Comments
|
||
|
1. Summary |
|
|
|
Thank you very much for taking the time to sincerely review our manuscript. We appreciate your constructive feedback and the opportunity to improve our work. Please find our detailed responses below, with corresponding revisions clearly highlighted in the re-submitted files.
|
||
|
2. Questions for General Evaluation |
Reviewer’s Evaluation |
Response and Revisions |
|
Does the introduction provide sufficient background and include all relevant references? |
Can be improved |
|
|
Is the research design appropriate? |
Can be improved |
|
|
Are the methods adequately described? |
Must be improved |
|
|
Are the results clearly presented? |
Must be improved |
|
|
Are the conclusions supported by the results? |
Must be improved |
|
|
3. Point-by-point response to Comments and Suggestions for Authors |
||
|
Comments 1: The purpose of this study was to assess the relationship between 30-day all-cause mortality in sepsis patients and the CALLY index. Adult patients who received a diagnosis of sepsis in the emergency room between January 1, 2022, and January 1, 2025 were included in this retrospective cohort study. The CALLY index was computed. The 30-day all-cause mortality was the main outcome, and decision curve analysis, logistic regression, and XGBoost machine learning models were used for statistical analyses. Of the 1,644 patients who were included, 345 (21.0%) passed away within 30 days. Compared to survivors, non-survivors had a significantly higher CALLY index (72.4 [23.3 - 190] vs. 24.3 [15.9 - 34.6], p < 0.001). Together with lactate, white blood cell count, and neutrophil count, the CALLY index was one of the best indicators of sepsis-related mortality in machine learning-based feature selection. The CALLY index showed the highest net benefit in the 0.10–0.15 risk threshold range, according to decision curve analysis, indicating that it may be useful in clinical decision-making. The paper needs several improvements, especially regarding the methodology and results section. Please address the following comments.
|
||
|
Response 1: Thank you for your time and thoughtful feedback. We appreciate your summary of the study and have addressed the methodological and results-related suggestions in detail in the corresponding comments below.
|
||
|
Comments 2: In the abstract, Please specify which feature selection method was used. |
||
|
Response 2: Thank you for your suggestion. In the Methods section of the abstract, we have specified that four feature selection methods—gain score, SHAP values, Boruta, and LASSO regression—were used to evaluate predictor consistency.
Comments 3: In the abstract, Please add numerical findings of the machine learning models. Response 3: Thank you for the comment. We have revised the Results section of the abstract to include the numerical performance metrics of the machine learning models.
Among the five machine learning models the XGBoost model achieved the highest performance (AUC: 0.995, R²: 0.867, MAE: 0.063, RMSE: 0.145). In gain-based feature selection, the CALLY index emerged as the top predictor (gain: 0.187), followed by serum lactate (0.185) and white blood cell count (0.117). The CALLY index also ranked second in SHAP analysis (mean value: 0.317) and first in Boruta importance (mean importance: 37.54). DCA showed the highest net clinical benefit of the CALLY index within the 0.10–0.15 risk threshold range.
Comments 4: Please mention the novelty of the study at the end of the introduction. Response 4: Thank you for your valuable suggestion. In response to this comment, we have added a sentence at the end of the introduction to highlight the novelty of our study. This study uniquely integrates a composite biomarker reflecting inflammation, nutrition, and immunity into multiple machine learning frameworks to predict 30-day mortality in sepsis. Its methodological design allows both performance comparison across algorithms and evaluation of clinical utility, offering a data-driven perspective on early risk strati-fication in this high-risk population.
Comments 5: Please add a paragraph describing the paper's organization at the end of the introduction. Response 5: Thank you for the suggestion. In accordance with your recommendation, we have added a concluding paragraph to the Introduction section that outlines the structure and key components of the paper. The updated paragraph reads: To explore this association, the study compares clinical and laboratory characteristics between survivors and non-survivors, and develops multiple machine learning mod-els—including extreme gradient boosting, multilayer perceptron, random forest, support vector machine, and generalized linear model—to predict 30-day mortality. Different feature selection strategies are applied to assess the consistency of important predictors, and the clinical utility of the CALLY index is further evaluated using decision curve analysis.
Comments 6: The methodology section needs some details about the methods used. Please justify the rationale for choosing XGBoost, a gradient boosting decision tree algorithm. Response 6: We thank the reviewer for this valuable comment. In response, we have expanded the Methods section to include a justification for the selection of XGBoost, along with a brief explanation of the rationale for using other machine learning models. Specifically, we clarified that XGBoost was chosen due to its proven performance in previous sepsis studies, particularly in handling nonlinear interactions, high-dimensional data, and missing values. XGBoost's regularization capabilities and interpretability through gain-based feature importance metrics also made it well-suited for our study objectives. We have also explained the inclusion of four additional models—MLP, random forest, SVM, and GLM—to enable a comparative analysis across different algorithmic classes. This addition can now be found in the Machine Learning Modeling subsection of the Methods section. To construct predictive models for 30-day mortality in sepsis, five supervised machine learning algorithms were implemented: extreme gradient boosting (XGBoost), multilayer perceptron (MLP), random forest, support vector machine (SVM), and generalized linear model (GLM). These algorithms were selected to encompass a diverse range of model architectures—boosted decision trees, neural networks, ensemble bagging, kernel-based classifiers, and linear models—to enable a comprehensive comparison across different methodological frameworks. XGBoost was included due to its demonstrated superiority in sepsis prognostication, particularly in handling missing data, capturing nonlinear rela-tionships, and minimizing overfitting through built-in regularization. Prior studies have shown that XGBoost effectively models complex clinical datasets and achieves strong discriminative performance in sepsis-related mortality prediction [15,16]. MLP was se-lected as a deep learning approach capable of capturing higher-order interactions that may not be linearly separable. Although more sensitive to overfitting, its inclusion provided insight into the behavior of multilayer neural networks in this context. Random forest was chosen for its robustness and inherent feature importance estimation, which has been previously leveraged in sepsis datasets for variable ranking and prediction [17]. SVM was incorporated as a classical algorithm known to perform well in high-dimensional spaces, especially when margin-based separation is feasible. Finally, GLM was included as a transparent, interpretable model commonly used in biomedical research, serving as a comparative benchmark against more complex models. The inclusion of these five algorithms allowed for evaluation of the relative advantages and limitations of each modeling approach in sepsis-related mortality prediction, while minimizing algorithmic bias. Subsequent comparisons of predictive performance and feature selection outcomes were conducted to guide model selection.
Comments 7: Please also explain the rationale for choosing gain metric for feature selection. Response 7: Thank you for pointing this out. Initially, we selected the gain metric for feature selection because of its direct alignment with XGBoost’s internal tree-splitting logic, computational efficiency, and ease of interpretation. These properties made it a practical starting point for identifying impactful features within our dataset. However, as suggested in your subsequent comment (see Comment 8), we expanded our analysis to include three additional feature selection methods—SHAP values, Boruta, and LASSO—and trained separate XGBoost models using the feature subsets identified by each method. These approaches were chosen to represent different classes of feature selection strategies: model-based (gain and SHAP), wrapper-based (Boruta), and regularization-based (LASSO). We were pleased to observe that the CALLY index emerged as a top-ranking predictor in three out of the four approaches, reinforcing its prognostic relevance. Since we incorporated multiple independent methods, we did not elaborate further on why gain was initially selected, as the overall analysis no longer depends on a single method. We appreciate your insightful suggestion, which helped improve the depth and robustness of the study.
Comments 8: Please use more feature selection approaches and compare their performance. Response 8: Thank you for this valuable suggestion. In response, we incorporated and compared three additional feature selection techniques: SHAP (SHapley Additive exPlanations) values, Boruta, and LASSO (Least Absolute Shrinkage and Selection Operator), alongside the original gain-based approach. Each method generated a distinct set of top features, summarized in the updated Table 3 and Figure 1. Importantly, the CALLY index emerged as a top-ranking feature in three out of the four methods—ranked 1st by gain, 2nd by SHAP, and 1st by Boruta—but was excluded by LASSO, likely due to its composite nature and multicollinearity penalties.
To evaluate the practical impact of feature selection strategy, we trained four separate XGBoost models using the selected features from each method and compared their performance using metrics including AUC, R², MAE, and RMSE. Additionally, DeLong’s test was used to statistically compare AUC values between models. No statistically significant differences were observed among any pairwise comparisons (all p > 0.05), indicating comparable discrimination performance. These additions and comparisons have been incorporated into the revised Results section (see below for details). The Results section now includes a full comparison of four XGBoost models trained on features selected by gain score, SHAP values, Boruta, and LASSO. This includes exact AUC values with 95% confidence intervals and statistical comparisons using DeLong’s test. The updated paragraph reads: We trained four distinct XGBoost models using features selected by different methods: gain score, SHAP values, Boruta, and LASSO. The top-ranked features identified by each method are summarized in Table 3, demonstrating considerable overlap across approaches (Figure 1). Notably, the CALLY index consistently ranked among the most influential predictors—first in the gain-based model, second in the SHAP-based model, and the top feature in Boruta's importance scores—while it was excluded by LASSO due to its penalization of multicollinearity and composite features. To evaluate the impact of feature selection strategy, we trained and tested models using each subset of selected features. The gain-based model achieved an AUC of 0.99517 (95% CI: 0.99071–0.99964), with R² = 0.867, MAE = 0.063, and RMSE = 0.145 on the test dataset. The SHAP-based model achieved an AUC of 0.99712 (95% CI: 0.99468–0.99956), with R² = 0.879, MAE = 0.063, and RMSE = 0.140. The Boruta-based model yielded an AUC of 0.99843 (95% CI: 0.997–1.000), with R² = 0.904, MAE = 0.059, and RMSE = 0.128. The LASSO-based model achieved an AUC of 0.99723 (95% CI: 0.995–1.000), with R² = 0.881, MAE = 0.066, and RMSE = 0.141. To statistically compare the discrimination performance of these models, we applied DeLong’s test for correlated ROC curves. There was no statistically significant difference in AUC between the gain- and SHAP-based models (p = 0.1675), gain and Boruta (p = 0.0865), or gain and LASSO (p = 0.3288). Similarly, the SHAP-based model did not differ significantly from Boruta (p = 0.1043) or LASSO (p = 0.9326). The Boruta- and LASSO-based models also did not differ significantly (p = 0.2480).
Comments 9: In addition, use more machine learning models. Response 9: We thank the reviewer for this valuable suggestion. In addition to the original model, we included four additional machine learning algorithms: multilayer perceptron, random forest, support vector machine, and generalized linear model. The updated paragraph reads: To assess the prognostic utility of the CALLY index alongside other clinical parameters, we developed multiple machine learning models to predict 30-day sepsis-related mortality. Specifically, we compared the performance of five different algorithms: extreme gradient boosting (XGBoost), multilayer perceptron (MLP), random forest, support vector machine (SVM), and generalized linear model (GLM). Each model was trained and evaluated using the same dataset, and their predictive performance was assessed on the test set using metrics including the area under the ROC curve (AUC), R², mean absolute error (MAE), and root mean square error (RMSE). Summary metrics are presented in Table 3. The gain-based XGBoost model achieved the highest test performance among all machine learning models, with an AUC of 0.995 (95% CI: 0.991–1.000), R² of 0.867, MAE of 0.063, and RMSE of 0.145 (Table 3). To evaluate its statistical superiority, we compared it against MLP, random forest, SVM, and GLM using DeLong’s test. XGBoost significantly outperformed all other models in terms of AUC (p < 0.001 for each pairwise comparison), confirming its superior discriminative power for sepsis-related mortality prediction.
Comments 10: How was data split for training and testing the machine learning models? Response 10: Thank you for this important question. The dataset was randomly divided into training (80%) and testing (20%) subsets, stratified by 30-day mortality to preserve class balance. All model training, hyperparameter tuning, and feature selection procedures were conducted exclusively on the training set to prevent data leakage and ensure generalizability. This information has now been clarified in the revised Methods section. The updated paragraph reads: The dataset was randomly divided into training (80%) and testing (20%) subsets, stratified by mortality outcome to preserve class balance. All model training, hyperparameter tuning, and performance evaluation were conducted on the training dataset, with test performance assessed independently to avoid data leakage.
|
||
|
4. Response to Comments on the Quality of English Language |
||
|
Point 1: The English is fine and does not require any improvement. |
||
|
Response 1: Thank you |
||
Reviewer 2 Report
Comments and Suggestions for Authors
The authors prepared the article very well.
The introduction is a proper introduction to the subject. The discussion does not raise any objections. It is appropriately conducted, and the authors refer correctly to existing studies. The conclusions are correct, and the results of the studies are presented.
I have a few minor comments
1. The authors unnecessarily explain the abbreviation CRP several times. One explanation is enough.
2. In Table 2, there is a technical error in the recording of the number of survivors and deceased. This should be corrected.
3. A diagram of the study group would be helpful, with information on how many people were considered, how many were excluded, and how many finally completed the study.
4. To increase readability - I suggest that the authors prepare a figure presenting the most essential features of the study along with the results.
The discussion does not raise my objections. It is appropriately conducted, and the authors refer correctly to existing studies. The conclusions are formulated correctly.
Author Response
For research article
|
Response to Reviewer 2 Comments
|
||
|
1. Summary |
|
|
|
Thank you very much for taking the time to sincerely review our manuscript. We appreciate your constructive feedback and the opportunity to improve our work. Please find our detailed responses below, with corresponding revisions clearly highlighted in the re-submitted files.
|
||
|
2. Questions for General Evaluation |
Reviewer’s Evaluation |
Response and Revisions |
|
Does the introduction provide sufficient background and include all relevant references? |
Can be improved |
|
|
Is the research design appropriate? |
Yes |
|
|
Are the methods adequately described? |
Can be improved |
|
|
Are the results clearly presented? |
Can be improved |
|
|
Are the conclusions supported by the results? |
Yes |
|
|
3. Point-by-point response to Comments and Suggestions for Authors |
||
|
Comments 1: The authors prepared the article very well. The introduction is a proper introduction to the subject. The discussion does not raise any objections. It is appropriately conducted, and the authors refer correctly to existing studies. The conclusions are correct, and the results of the studies are presented. |
||
|
Response 1: We sincerely thank the reviewer for their positive evaluation and thoughtful feedback regarding the introduction, discussion, and conclusions. We are grateful for the acknowledgment and are pleased that these sections were found to be clear and well-prepared.
|
||
|
Comments 2: In Table 2, there is a technical error in the recording of the number of survivors and deceased. This should be corrected. Response 2: We thank the reviewer for noting this. The error in the survivor and deceased group labeling in Table 2 has been carefully reviewed and corrected in the revised version of the manuscript.
Comments 3: A diagram of the study group would be helpful, with information on how many people were considered, how many were excluded, and how many finally completed the study. Response 3: Thank you for the helpful suggestion. As recommended, we have added a patient flow diagram summarizing the number of patients screened, the reasons for exclusion, and the final number included in the analysis. This has been added as Figure 1 in the revised manuscript.
Comments 4: To increase readability - I suggest that the authors prepare a figure presenting the most essential features of the study along with the results. Response 4: We appreciate this insightful suggestion. In response, we have created a summary visual that presents the key design elements of the study, the CALLY index formula, and an overview of the machine learning-based results. This has been included as graphical abstract in the revised manuscript to enhance reader comprehension.
Comments 5: The discussion does not raise my objections. It is appropriately conducted, and the authors refer correctly to existing studies. The conclusions are formulated correctly. Response 5: We are grateful for your kind assessment and are pleased that the discussion and conclusions were found appropriate and well-referenced. Thank you again for your supportive and constructive review.
|
||
|
4. Response to Comments on the Quality of English Language |
||
|
Point 1: The English is fine and does not require any improvement. |
||
|
Response 1: Thank you |
||
Round 2
Reviewer 1 Report
Comments and Suggestions for Authors
The authors have addressed my comments